# Visual Reference Resolution using Attention Memory for Visual Dialog

**Paul Hongsuck Seo**[†]     **Andreas Lehrmann**[§]     **Bohyung Han**[†]     **Leonid Sigal**[§]
[†]POSTECH                                    [§]Disney Research
{hsseo, bhhan}@postech.ac.kr  {andreas.lehrmann, lsigal}@disneyresearch.com

## Abstract

Visual dialog is a task of answering a series of inter-dependent questions given an input image, and often requires to resolve visual references among the questions. This problem is different from visual question answering (VQA), which relies on spatial attention (*a.k.a. visual grounding*) estimated from an image and question pair. We propose a novel attention mechanism that exploits visual attentions in the past to resolve the current reference in the visual dialog scenario. The proposed model is equipped with an associative attention memory storing a sequence of previous (attention, key) pairs. From this memory, the model retrieves the previous attention, taking into account recency, which is most relevant for the current question, in order to resolve potentially ambiguous references. The model then merges the retrieved attention with a tentative one to obtain the final attention for the current question; specifically, we use dynamic parameter prediction to combine the two attentions conditioned on the question. Through extensive experiments on a new synthetic visual dialog dataset, we show that our model significantly outperforms the state-of-the-art (by $\approx 16$ % points) in situations, where visual reference resolution plays an important role. Moreover, the proposed model achieves superior performance ($\approx 2$ % points improvement) in the Visual Dialog dataset [1], despite having significantly fewer parameters than the baselines.

## 1   Introduction

In recent years, advances in the design and optimization of deep neural network architectures have led to tremendous progress across many areas of computer vision (CV) and natural language processing (NLP). This progress, in turn, has enabled a variety of multi-modal applications spanning both domains, including image captioning [2–4], language grounding [5, 6], image generation from captions [7, 8], and visual question answering (VQA) on images [9–21] and videos [22–24].

The VQA task, in particular, has received broad attention because its formulation requires a universal understanding of image content. Most state-of-the-art methods [10, 13, 15] address this inherently challenging problem through an attention mechanism [3] that allows to visually ground linguistic expressions; they identify the region of visual interest referred to by the question and predict the answer based on the visual information in that region.

More recently, Visual Dialog [1] has been introduced as a generalization of the VQA task. Unlike VQA, where every question is asked independently, a visual dialog system needs to answer a *sequence* of questions about an input image. The sequential and inter-dependent property of questions in a dialog presents additional challenges. Consider the simple image and partial dialog in Figure 1. Some questions (*e.g.*, #1: 'How many 9's are there in the image?') contain the full information needed to attend to the regions within the image and answer the question accurately. Other questions (*e.g.*, #6: 'What is the number of the blue digit?') are ambiguous on their own and require knowledge obtained from the prior questions (1, 2, 3, and 5 in particular) in order to resolve attention to the specific region

| # | Question | Answer |
|---|----------|--------|
| 1 | How many 9's are there in the image? | four |
| 2 | How many brown digits are there among them? | one |
| 3 | What is the background color of the digit at the left of it? | white |
| 4 | What is the style of the digit? | flat |
| 5 | What is the color of the digit at the left of it? | blue |
| 6 | What is the number of the blue digit? | 4 |
| 7 | Are there other blue digits? | two |

Figure 1: **Example from MNIST Dialog.** Each pair consists of an image (left) and a set of sequential questions with answers (right).

the expression ('the blue digit') is referring to. This process of *visual reference resolution*[1] is the key component required to localize attention accurately in the presence of ambiguous expressions and thus plays a crucial role in extending VQA approaches to the visual dialog task.

We perform visual reference resolution relying on a novel attention mechanism that employs an associative memory to obtain a visual reference for an ambiguous expression. The proposed model utilizes two types of intermediate attentions: tentative and retrieved ones. The *tentative attention* is calculated solely based on the current question (and, optionally, the dialog history), and is capable of focusing on an appropriate region when the question is unambiguous. The *retrieved attention*, used for visual reference resolution, is the most relevant previous attention available in the associative memory. The final attention for the current question is obtained by combining the two attention maps conditioned on the question; this is similar to neural module networks [12, 14], which dynamically combine discrete attention modules, based on a question, to produce the final attention. For this task, our model adopts a dynamic parameter layer [9] that allows us to work with continuous space of dynamic parametrizations, as opposed to the discrete set of parametrizations in [12, 14].

**Contributions**   We make the following contributions. (1) We introduce a novel attention process that, in addition to direct attention, resolves visual references by modeling the sequential dependency of the current question on previous attentions through an associative attention memory; (2) We perform a comprehensive analysis of the capacity of our model for the visual reference resolution task using a synthetic visual dialog dataset (MNIST dialog) and obtain superior performance compared to all baseline models. (3) We test the proposed model in a visual dialog benchmark (VisDial [1]) and show state-of-the-art performance with significantly fewer parameters.

## 2   Related Work

**Visual Dialog**   Visual dialogs were recently proposed in [1] and [25], focusing on different aspects of a dialog. While the conversations in the former contain free-form questions about arbitrary objects, the dialogs in the latter aim at object discovery through a series of yes/no questions. Reinforcement learning (RL) techniques were built upon those works in [26] and [27]. Das et al. [26] train two agents by playing image guessing games and show that they establish their own communication protocol and style of speech. In [27], RL is directly used to improve the performance of agents in terms of the task completion rate of goal-oriented dialogs. However, the importance of previous references has not yet been explored in the visual dialog task.

**Attention for Visual Reference Resolution**   While visual dialog is a recent task, VQA has been studied extensively and attention models have been known to be beneficial for answering independent questions [10–16]. However, none of those methods incorporate visual reference resolution, which is neither necessary nor possible in VQA but essential in visual dialog. Beyond VQA, attention models are used to find visual groundings of linguistic expressions in a variety of other multi-modal tasks, such as image captioning [3, 4], VQA in videos [22], and visual attributes prediction [28]. Common to most of these works, an attention is obtained from a single embedding of all linguistic inputs. Instead, we propose a model that embeds each question in a dialog separately and calculates the current question's attention by resolving its sequential dependencies through an attention memory and a dynamic attention combination process. We calculate an attention through a dynamic composition

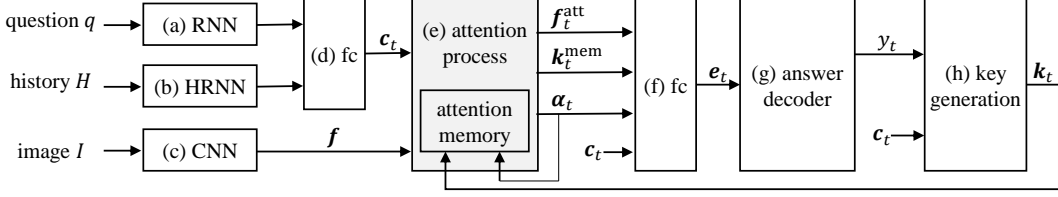

Figure 2: **Architecture of the proposed network.** The gray box represents the proposed attention process. Refer to Section 3 for the detailed description about individual modules (a)-(f).

process taking advantage of a question's semantic structure, which is similar to [12] and [14]. However, the proposed method still differs in that our attention process is designed to deal with ambiguous expressions in dialogs by dynamically analyzing the dependencies of questions at each time step. In contrast, [12] and [14] obtain the attention for a question based on its compositional semantics that is completely given at the time of the network structure prediction.

**Memory for Question Answering** Another line of closely related works is the use of a memory component to question answering models. Memory networks with end-to-end training are first introduced in [29], extending the original memory network [30]. The memories in these works are used to store some factoids in a given story and the supporting facts for answering questions are selectively retrieved through memory addressing. A memory network with an episodic memory was proposed in [31] and applied to VQA by storing the features at different locations of the memory [32]. While these memories use the contents themselves for addressing, [33] proposes associative memories that have a key-value pair at each entry and use the keys for addressing the value to be retrieved. Finally, the memory component is also utilized for visual dialog in [1] to actively select the previous question in the history. Memories in these previous memory networks store given factoids to retrieve a supporting fact. In contrast, our attention memory stores previous attentions, which represent grounded references for previous questions, to resolve the current reference based on the sequential dependency of the referring expressions. Moreover, we adopt an associative memory to use the semantics of QA pairs for addressing.

## 3    Visual Dialog Model with Attention Memory-based Reference Resolution

Visual dialog is the task of building an agent capable of answering a sequence of questions presented in the form of a dialog. Formally, we need to predict an answer $y_t \in \mathcal{Y}$, where $\mathcal{Y}$ is a set of discrete answers or a set of natural language phrases/sentences, at time $t$ given input image $I$, current question $q_t$, and dialog history $H = \{h_\tau | \ h_\tau = (q_\tau, y_\tau), \ 0 \le \tau < t\}$.

We utilize the encoder-decoder architecture recently introduced in [1], which is illustrated in Figure 2. Specifically, we represent a triplet $(q, H, I)$ with $\boldsymbol{e}_t$ by applying three different encoders, based on recurrent (RNN with long-short term memory units), hierarchical recurrent (HRNN)[2] and convolutional (CNN) neural networks, followed by attention and fusion units (Figure 2 (a)-(f)). Our model then decodes the answer $y_t$ from the encoded representation $\boldsymbol{e}_t$ (Figure 2 (g)). Note that, to obtain the encoded representation $\boldsymbol{e}_t$, the CNN image feature map $\boldsymbol{f}$ computed from $I$ undergoes a soft spatial attention process guided by the combination of $q_t$ and $H$ as follows:

$$\boldsymbol{c}_t = \texttt{fc}(\text{RNN}(q_t), \text{HRNN}(H)) \tag{1}$$

$$\boldsymbol{f}_t^{\text{att}} = [\boldsymbol{\alpha}_t(\boldsymbol{c}_t)]^\top \cdot \boldsymbol{f} = \sum_{n=1}^{N} \alpha_{t,n}(\boldsymbol{c}_t) \cdot \boldsymbol{f}_n, \tag{2}$$

where $\texttt{fc}$ (Figure 2 (d)) denotes a fully connected layer, $\boldsymbol{\alpha}_n(\boldsymbol{c}_t)$ is the attention map conditioned on a fused encoding of $q_t$ and $H$, $n$ is the location index in the feature map, and $N$ is the size of the spatial grid of the feature map. This attention mechanism is the critical component that allows the decoder to focus on relevant regions of the input image; it is also the main focus of this paper.

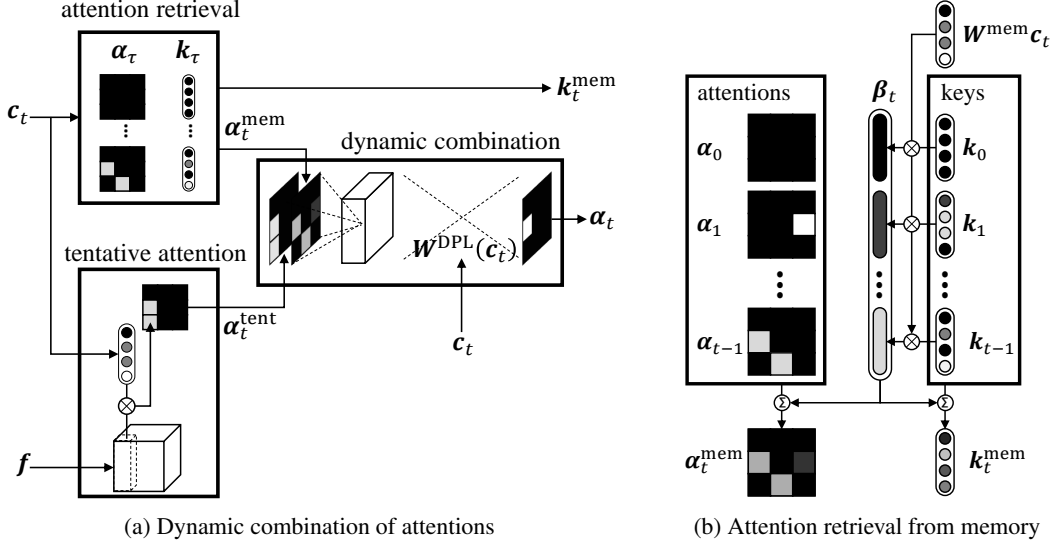

(a) Dynamic combination of attentions      (b) Attention retrieval from memory

Figure 3: **Attention process for visual dialog task.** (a) The tentative and relevant attentions are first obtained independently and then dynamically combined depending on the question embedding. (b) Two boxes represent memory containing attentions and corresponding keys. Question embedding $c_t$ is projected by $W^{\text{mem}}$ and compared with keys using inner products, denoted by crossed circles, to generate address vector $\beta_t$. The address vector is then used as weights for computing a weighted average of all memory entries, denoted by $\Sigma$ within circle, to retrieve memory entry $(\alpha_t^{\text{mem}}, k_t^{\text{mem}})$.

We make the observation that, for certain questions, attention can be resolved directly from $c_t$. This is called *tentative attention* and denoted by $\alpha_t^{\text{tent}}$. This works well for questions like #1 in Figure 1, which are free from dialog referencing. For other questions like #6, resolving reference linguistically would be difficult (*e.g.*, linguistic resolution may look like: 'What number of the digit to the left to the left of the brown 9'). That said, #6 is straightforward to answer if the attention utilized to answer #5 is retrieved. This process of visual reference resolution gives rise to attention retrieval $\alpha_t^{\text{mem}}$ from the memory. The final attention $\alpha_t(c_t)$ is computed using dynamic parameter layer, where the parameters are conditioned on $c_t$. To summarize, an attention is composed of three steps in the proposed model: tentative attention, relevant attention retrieval, and dynamic attention fusion as illustrated in Figure 3a. We describe the details of each step below.

## 3.1 Tentative Attention

We calculate the tentative attention by computing similarity, in the joint embedding space, of the encoding of the question and history, $c_t$, and each feature vector, $f_n$, in the image feature grid $f$:

$$s_{t,n} = \left(\mathbf{W}_c^{\text{tent}} c_t\right)^\top \left(\mathbf{W}_f^{\text{tent}} f_n\right) \tag{3}$$

$$\alpha_t^{\text{tent}} = \text{softmax}\left(\{s_{t,n}, 1 < n < N\}\right), \tag{4}$$

where $\mathbf{W}_c^{\text{tent}}$ and $\mathbf{W}_f^{\text{tent}}$ are projection matrices for the question and history encoding and the image feature vector, respectively, and $s_{t,n}$ is an attention score for a feature at the spatial location $n$.

## 3.2 Relevant Attention Retrieval from Attention Memory

As a reminder, in addition to the tentative attention, our model obtains the most relevant previous attention using an attention memory for visual reference resolution.

**Associative Attention Memory** The proposed model is equipped with an associative memory, called an attention memory, to store previous attentions. The attention memory $M_t = \{(\alpha_0, k_0), (\alpha_1, k_1), \ldots, (\alpha_{t-1}, k_{t-1})\}$ stores all the previous attention maps $\alpha_\tau$ with their corresponding keys $k_\tau$ for associative addressing. Note that $\alpha_0$ is NULL attention and set to all zeros. The NULL attention can be used when no previous attention reference is required for the current reference resolution.

The most relevant previous attention is retrieved based on the key comparison as illustrated in Figure 3b. Formally, the proposed model addresses the memory given the embedding of the current question and history $c_t$ using

$$m_{t,\tau} = (\boldsymbol{W}^{\mathrm{mem}} \boldsymbol{c}_t)^\top \boldsymbol{k}_\tau \quad \text{and} \quad \boldsymbol{\beta}_t = \mathrm{softmax}\left(\{m_{t,\tau}, 0 < \tau < t - 1\}\right), \tag{5}$$

where $\boldsymbol{W}^{\mathrm{mem}}$ projects the question and history encoding onto the semantic space of the memory keys. The relevant attention $\boldsymbol{\alpha}_t^{\mathrm{mem}}$ and key $\boldsymbol{k}_t^{\mathrm{mem}}$ are then retrieved from the attention memory using the computed addressing vector $\boldsymbol{\beta}_t$ by

$$\boldsymbol{\alpha}_t^{\mathrm{mem}} = \sum_{\tau=0}^{t-1} \boldsymbol{\beta}_{t,\tau} \boldsymbol{\alpha}_\tau \quad \text{and} \quad \boldsymbol{k}_t^{\mathrm{mem}} = \sum_{\tau=0}^{t-1} \boldsymbol{\beta}_{t,\tau} \boldsymbol{k}_\tau. \tag{6}$$

This relevant attention retrieval allows the proposed model to resolve the visual reference by indirectly resolving coreferences [34–36] through the memory addressing process.

**Incorporating Sequential Dialog Structure** While the associative addressing is effective in retrieving the most relative attention based on the question semantics, we can improve the performance by incorporating sequential structure of the questions in a dialog. Considering that more recent attentions are more likely to be referred again, we add an extra term to Eq. (5) that allows preference for sequential addressing, *i.e.*, $m'_{t,\tau} = (\boldsymbol{W}^{\mathrm{mem}} \boldsymbol{c}_t)^\top \boldsymbol{k}_\tau + \theta(t - \tau)$ where $\theta$ is a learnable parameter weighting the relative time distance $(t - \tau)$ from the current time step.

## 3.3 Dynamic Attention Combination

After obtaining both attentions, the proposed model combines them. The two attention maps $\boldsymbol{\alpha}_t^{\mathrm{tent}}$ and $\boldsymbol{\alpha}_t^{\mathrm{mem}}$ are first stacked and fed to a convolution layer to locally combine the attentions. After generating the locally combined attention features, it is flattened and fed to a fully connected (`fc`) layer with softmax generating the final attention map. However, a `fc` layer with fixed weights would always result in the same type of combination although the merging process should, as we argued previously, depend on the question. Therefore, we adopt the dynamic parameter layer introduced in [9] to adapt the weights of the `fc` layer conditioned on the question at test time. Formally, the final attention map $\boldsymbol{\alpha}_t(\boldsymbol{c}_t)$ for time $t$ is obtained by

$$\boldsymbol{\alpha}_t(\boldsymbol{c}_t) = \mathrm{softmax}\left(\boldsymbol{W}^{\mathrm{DPL}}(\boldsymbol{c}_t) \cdot \gamma(\boldsymbol{\alpha}_t^{\mathrm{tent}}, \boldsymbol{\alpha}_t^{\mathrm{mem}})\right), \tag{7}$$

where $\boldsymbol{W}^{\mathrm{DPL}}(\boldsymbol{c}_t)$ are the dynamically determined weights and $\gamma(\boldsymbol{\alpha}_t^{\mathrm{tent}}, \boldsymbol{\alpha}_t^{\mathrm{mem}})$ is the flattened output of the convolution obtained from the stacked attention maps. As in [9], we use a hashing technique to predict the dynamic parameters without explosive increase of network size.

## 3.4 Additional Components and Implementation

In addition to the attended image feature, we find other information useful for answering the question. Therefore, for the final encoding $\boldsymbol{e}_t$ at time step $t$, we fuse the attended image feature embedding $\boldsymbol{f}_t^{\mathrm{att}}$ with the context embedding $\boldsymbol{c}_t$, the attention map $\boldsymbol{\alpha}_t$ and the retrieved key $\boldsymbol{k}_t^{\mathrm{mem}}$ from the memory, by a `fc` layer after concatenation (Figure 2f).

Finally, when we described the associative memory in Section 3, we did not specify the memory key generation procedure. In particular, after answering the current question, we append the computed attention map to the memory. When storing the current attention into memory, the proposed model generates a key $\boldsymbol{k}_t$ by fusing the context embedding $\boldsymbol{c}_t$ with the current answer embedding $\boldsymbol{a}_t$ through a `fc` layer (Figure 2h). Note that an answer embedding $\boldsymbol{a}_t$ is obtained using LSTM.

**Learning** Since all the modules of the proposed network are fully differentiable, the entire network can be trained end-to-end by standard gradient-based learning algorithms.

## 4 Experiments

We conduct two sets of experiments to verify the proposed model. To highlight the model's ability to resolve visual references, we first perform experiment with a synthetic dataset that is explicitly designed to contain ambiguous expressions and strong inter-dependency among questions in the visual dialog. We then show that the model also works well in the real VisDial [1] benchmark.

| Basemodel | +H | +SEQ | Accuracy |
|---|---|---|---|
| I | – | – | 20.18 |
| Q | – | – | 36.58 |
|  | ✓ | – | 37.58 |
| LF [1] | ✓ | – | 45.06 |
| HRE [1] | ✓ | – | 49.10 |
| MN [1] | ✓ | – | 48.51 |
| ATT | – | – | 62.62 |
|  | ✓ | – | 79.72 |
| AMEM | – | – | 87.53 |
|  | ✓ | – | 89.20 |
|  | – | ✓ | 90.05 |
|  | ✓ | ✓ | **96.39** |

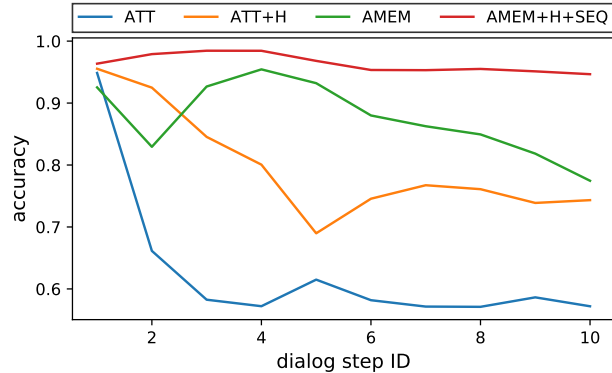

Figure 4: **Results on MNIST Dialog.** Answer prediction accuracy [%] of all models for all questions (left) and accuracy curves of four models at different dialog steps (right). +H and +SEQ represent the use of history embeddings in models and addressing with sequential preference, respectively.

## 4.1 MNIST Dialog Dataset

**Experimental Setting**  We create a synthetic dataset, called MNIST Dialog[3], which is designed for the analysis of models in the task of visual reference resolution with ambiguous expressions. Each image in MNIST Dialog contains a $4 \times 4$ grid of MNIST digits and each MNIST digit in the grid has four randomly sampled attributes, *i.e.*, $\mathrm{color} = \{\mathrm{red, blue, green, purple, brown}\}$, $\mathrm{bgcolor} = \{\mathrm{cyan, yellow, white, silver, salmon}\}$, $\mathrm{number} = \{x | 0 \le x \le 9\}$ and $\mathrm{style} = \{\mathrm{flat, stroke}\}$, as illustrated in Figure 1. Given the generated image from MNIST Dialog, we automatically generate questions and answers about a subset of the digits in the grid that focus on visual reference resolution. There are two types of questions: (i) counting questions and (ii) attribute questions that refer to a single target digit. During question generation, the target digits for a question is selected based on a subset of the previous targets referred to by ambiguous expressions, as shown in Figure 1. For ease of evaluation, we generate a single word answer rather than a sentence for each question and there are a total of 38 possible answers ($\frac{1}{38}$ chance performance). We generated 30K / 10K / 10K images for training / validation / testing, respectively, and three ten-question dialogs for each image.

The dimensionality of the word embedding and the hidden state in the LSTMs are set to 32 and 64, respectively. All LSTMs are single-layered. Since answers are single words, the answer embedding RNN is replaced with a word embedding layer in both the history embedding module and the memory key generation module. The image feature extraction module is formed by stacking four $3 \times 3$ convolutional layers with a subsequent $2 \times 2$ pooling layer. The first two convolutional layers have 32 channels, while there are 64 channels in the last two. Finally, we use 512 weight candidates to hash the dynamic parameters of the attention combination process. The entire network is trained end-to-end by minimizing the cross entropy of the predicted answer distribution at every step of the dialogs.

We compare our model (AMEM) with three different groups of baselines. The simple baselines show the results of using statistical priors, where answers are obtained using image (I) or question (Q) only. We also implement the late fusion model (LF), the hierarchical recurrent encoder with attention (HREA) and the memory network encoder (MN) introduced in [1]. Additionally, an attention-based model (ATT), which directly uses tentative attention, without memory access, is implemented as a strong baseline. For some models, two variants are implemented: one using history embeddings and the other one not. These variations give us insights on the effect of using history contexts and are distinguished by +H. Finally, another two versions of the proposed model, orthogonal to the previous ones, are implemented with and without the sequential preference in memory addressing (see above), which is denoted by +SEQ.

**Results**  Figure 4 shows the results on MNIST Dialog. The answer prediction accuracy over all questions of dialogs is presented in the table on the left. It is noticeable that the models using attention mechanisms (AMEM and ATT) significantly outperform the previous baseline models (LF, HRE and MN) introduced in [1], while these baselines still perform better than the simple baseline models. This signifies the importance of attention in answering questions, consistent with previous works [10–14].

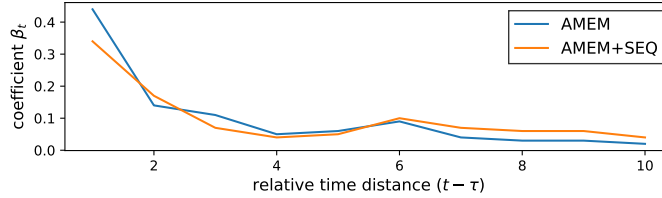

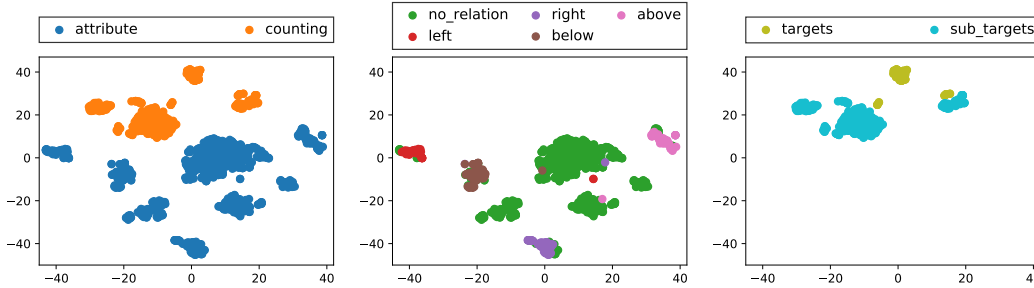

Figure 5: **Memory addressing coefficients with and without sequential preference.** Both models put large weights on recent elements (smaller relative time difference) to deal with the sequential structure of dialogs.

Figure 6: **Characteristics of dynamically predicted weights for attention combination.** Dynamic weights are computed from 1,500 random samples at dialog step 3 and plotted by t-SNE. Each figure presents clusters formed by different semantics of questions. (left) Clusters generated by different question types. (middle) Subclusters formed by types of spatial relationships in attribute questions. (right) Subclusters formed by ways of specifying targets in counting questions; cluster sub_targets contains questions whose current target digits are included in the targets of the previous question.

Extending ATT to incorporate history embeddings during attention map estimation increases the accuracy by about 17%, resulting in a strong baseline model.

However, even the simplest version of the proposed model, which does not use history embeddings or addressing with sequential preference, already outperforms the strong baseline by a large margin. Note that this model still has indirect access to the history through the attention memory, although it does not have direct access to the encodings of past question/answer pairs when computing the attention. This signifies that the use of the attention memory is more helpful in resolving the current reference (and computing attention), compared to a method that uses more traditional tentative attention informed by the history encoding. Moreover, the proposed model with history embeddings further increases the accuracy by 1.7%. The proposed model reaches >96% accuracy when the sequential structure of dialogs is taken into account by the sequential preference in memory addressing.

We also present the accuracies of the answers at each dialog step for four models that use attentions in Figure 4 (right). Notably, the accuracy of ATT drops very fast as the dialog progresses and reference resolution is needed. Adding history embeddings to the tentative attention calculation somewhat reduces the degradation. The use of the attention memory gives a very significant improvement, particularly at later steps in the dialog when complex reference resolution is needed.

**Parameter Analysis** When we observed the learned parameter $\theta$ for the sequential preference, it is consistently negative in all experiments; it means that all models prefer recent elements. A closer look at the addressing coefficients $\beta_t$ with and without the sequential preference reveals that both variants have a clear preferences for recent elements, as depicted in Figure 5. It is interesting that the case without the bias term shows a stronger preference for recent information, but its final accuracy is lower than the version with the bias term. It seems that $W^{\mathrm{mem}}$ without bias puts too much weight on recent elements, resulting in worse performance. Based on this observation, we learn $W^{\mathrm{mem}}$ and $\theta$ jointly to find better coefficients than $W^{\mathrm{mem}}$ alone.

The dynamically predicted weights form clusters with respect to the semantics of the input questions as illustrated in Figure 6, where 1,500 random samples at step 3 of dialogs are visualized using t-SNE. In Figure 6 (left), the two question types (attribute and counting) create distinct clusters. Each of

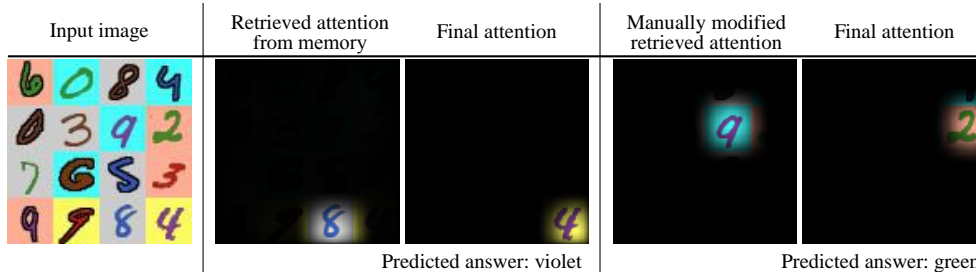

| History: | Are there any 9's in the image ? | three |
| | How many digits in a yellow background are there among them ? | one |
| | What is the color of the digit ? | red |
| | What is the color of the digit at the right of it ? | blue |
| | What is the style of the blue digit ? | flat |
| Current QA: | What is the color of the digit at the right of it ? | violet |

| Input image | Retrieved attention from memory | Final attention | Manually modified retrieved attention | Final attention |
|---|---|---|---|---|
| | | Predicted answer: violet | | Predicted answer: green |

Figure 7: **Qualitative analysis on MNIST Dialog.** Given an input image and a series of questions with their visual grounding history, we present the memory retrieved and final attentions for the current question in the second and third columns, respectively. The proposed network correctly attends to target reference and predicts correct answer. The last two columns present the manually modified attention and the final attention obtained from the modified attention, respectively. Experiment shows consistency of transformation between attentions and semantic interpretability of our model.

these, in turn, contains multiple sub-clusters formed by other semantics, as presented in Figure 6 (middle) and (right). In the cluster of attribute questions, sub-clusters are mainly made by types of spatial relationship used to specify the target digit (*e.g.*, #3 in Figure 1), whereas sub-clusters in counting questions are based on whether the target digits of the question are selected from the targets of the previous question or not (*e.g.*, #1 vs. #2 in Figure 1).

Figure 7 illustrates qualitative results. Based on the history of attentions stored in the attention memory, the proposed model retrieves the previous reference as presented in the second column. The final attention for the current question is then calculated by manipulating the retrieved attention based on the current question. For example, the current question in Figure 7 refers to the right digit of the previous reference, and the model identifies the target reference successfully (column 3) as the previous reference (column 2) is given accurately by the retrieved attention. To investigate consistency with respect to attention manipulation, we move the region of the retrieved attention manually (column 4) and observe the final attention map calculated from the modified attention (column 5). It is clear that our reference resolution procedure works consistently even with the manipulated attention and responds to the question accordingly. This shows a level of semantic interpretability of our model. See more qualitative results in Section A of our supplementary material.

## 4.2 Visual Dialog (VisDial) Dataset

**Experimental Setting**    In the VisDial [1] dataset[4], the dialogs are collected from MS-COCO [37] images and their captions. Each dialog is composed of an image, a caption, and a sequence of ten QA pairs. Unlike in MNIST Dialog, answers to questions in VisDial are in free form text. Since each dialog always starts with an initial caption annotated in MS-COCO, the initial history is always constructed using the caption. The dataset provides 100 answer candidates for each question and accuracy of a question is measured by the rank of the matching ground-truth answer. Note that this dataset is less focused on visual reference resolution and contains fewer ambiguous expressions compared to MNIST Dialog. We estimate the portion of questions containing ambiguous expressions to be 94% and 52% in MNIST Dial and VisDial, respectively[5].

While we compare our model with various encoders introduced in [1], we fix the decoder to a discriminative decoder that directly ranks the answer candidates through their embeddings. Our baselines include three visual dialog models, *i.e.*, late fusion model (LF), hierarchical recurrent encoder (HRE) and memory network encoder (MN), and two attention based VQA models (SAN and

Table 1: **Experimental results on VisDial.** We show the number of parameters, mean reciprocal rank (MRR), recall@$k$ and mean rank (MR). +H and ATT indicate use of history embeddings in prediction and attention mechanism, respectively.

| Model | +H | ATT | # of params | MRR | R@1 | R@5 | R@10 | MR |
|---|---|---|---|---|---|---|---|---|
| Answer prior [1] | – | – | n/a | 0.3735 | 23.55 | 48.52 | 53.23 | 26.50 |
| LF-Q [1] | – | – | 8.3 M (3.6x) | 0.5508 | 41.24 | 70.45 | 79.83 | 7.08 |
| LF-QH [1] | ✓ | – | 12.4 M (5.4x) | 0.5578 | 41.75 | 71.45 | 80.94 | 6.74 |
| LF-QI [1] | – | – | 10.4 M (4.6x) | 0.5759 | 43.33 | 74.27 | 83.68 | 5.87 |
| LF-QIH [1] | ✓ | – | 14.5 M (6.3x) | 0.5807 | 43.82 | 74.68 | 84.07 | 5.78 |
| HRE-QH [1] | ✓ | – | 15.0 M (6.5x) | 0.5695 | 42.70 | 73.25 | 82.97 | 6.11 |
| HRE-QIH [1] | ✓ | – | 16.8 M (7.3x) | 0.5846 | 44.67 | 74.50 | 84.22 | 5.72 |
| HREA-QIH [1] | ✓ | – | 16.8 M (7.3x) | 0.5868 | 44.82 | 74.81 | 84.36 | 5.66 |
| MN-QH [1] | ✓ | – | 12.4 M (5.4x) | 0.5849 | 44.03 | 75.26 | 84.49 | 5.68 |
| MN-QIH [1] | ✓ | – | 14.7 M (6.4x) | 0.5965 | 45.55 | 76.22 | 85.37 | 5.46 |
| SAN-QI [10] | – | ✓ | n/a | 0.5764 | 43.44 | 74.26 | 83.72 | 5.88 |
| HieCoAtt-QI [15] | – | ✓ | n/a | 0.5788 | 43.51 | 74.49 | 83.96 | 5.84 |
| AMEM-QI | – | ✓ | **1.7 M (0.7x)** | 0.6196 | 48.24 | 78.33 | 87.11 | 4.92 |
| AMEM-QIH | ✓ | ✓ | 2.3 M (1.0x) | 0.6192 | 48.05 | 78.39 | 87.12 | 4.88 |
| AMEM+SEQ-QI | – | ✓ | **1.7 M (0.7x)** | **0.6227** | **48.53** | **78.66** | **87.43** | **4.86** |
| AMEM+SEQ-QIH | ✓ | ✓ | 2.3 M (1.0x) | 0.6210 | 48.40 | 78.39 | 87.12 | 4.92 |

HieCoAtt) with the same decoder. The three visual dialog baselines are trained with different valid combinations of inputs, which are denoted by Q, I and H in the model names.

We perform the same ablation study of our model with the one for MNIST Dialog dataset. The `conv5` layer in VGG-16 [38] trained on ImageNet [39] is used to extract the image feature map. Similar to [1], all word embedding layers share their weights and an LSTM is used for embedding the current question. For the models with history embedding, we use additional LSTMs for the questions, the answers, and the captions in the history. Based on our empirical observation, we share the parameters of the question and caption LSTMs while having a separate set of weights for the answer LSTM. Every LSTM embedding sentences is two-layered, but the history LSTM of HRNN has a single layer. We employ 64 dimensional word embedding vectors and 128 dimensional hidden state for every LSTM. Note that the the dimensionality of our word embeddings and hidden state representations in LSTMs are significantly lower than the baselines (300 and 512 respectively). We train the network using Adam [40] with the initial learning rate of 0.001 and weight decaying factor 0.0001. Note that we do not update the feature extraction network based on VGG-16.

**Results** Table 1 presents mean reciprocal rank (MRR), mean rank (MR), and recall@$k$ of the models. Note that lower is better for MRs but higher is better for all other evaluation metrics. All variants of the proposed model outperform the baselines in all metrics, achieving the state-of-the-art performance. As observed in the experiments on MNIST Dialog, the models with sequential preference (+SEQ) show better performances compared to the ones without it. However, we do not see additional benefits from using a history embedding on VisDial, in contrast to MNIST Dialog. The proposed algorithm also has advantage over existing methods in terms of the number of parameters. Our full model only requires approximately 15% of parameters compared to the best baseline model without counting the parameters in the common feature extraction module based on VGG-16. In VisDial, the attention based VQA techniques with (near) state-of-the-art performances are not as good as the baseline models of [1] because they treat each question independently. The proposed model improves the performance on VisDial by facilitating the visual reference resolution process. Qualitative results for VisDial dataset are presented in Section B of the supplementary material.

# 5  Conclusion

We proposed a novel algorithm for answering questions in visual dialog. Our algorithm resolves visual references in dialog questions based on a new attention mechanism with an attention memory, where the model indirectly resolves coreferences of expressions through the attention retrieval process. We employ the dynamic parameter prediction technique to adaptively combine the tentative and retrieved attentions based on the question. We tested on both synthetic and real datasets and illustrated improvements.

**Acknowledgments**

This work was supported in part by the IITP grant funded by the Korea government (MSIT) [2017-0-01778, Development of Explainable Human-level Deep Machine Learning Inference Framework; 2017-0-01780, The Technology Development for Event Recognition/Relational Reasoning and Learning Knowledge based System for Video Understanding; 2016-0-00563, Research on Adaptive Machine Learning Technology Development for Intelligent Autonomous Digital Companion].

## Footnotes

[1]We coin this term by borrowing nomenclature, partially, from NLP, where coreference resolution attempts to solve the corresponding problem in language; the *visual* in *visual reference resolution* implies that we want to do both resolve and visually ground the reference used in the question.

[2]The questions and the answers of a history are independently embedded using LSTMs and then fused by a $\texttt{fc}$ layer with concatenation to form QA encodings. The fused QA embedding at each time step is finally fed to another LSTM and the final output is used for the history encoding.

[3]The dataset is available at `http://cvlab.postech.ac.kr/research/attmem`

[4]We use recently released VisDial v0.9 with the benchmark splits [1].

[5]We consider pronouns and definite noun phrases as ambiguous expressions and count them using a POS tagger in NLTK (http://www.nltk.org/).

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
