[Supplementary Material]

# Visual Reference Resolution using Attention Memory for Visual Dialog

*Supplementary Document*

## A More Qualitative Results on MNIST Dialog

| Input image | # | Questions | Answers |
|---|---|---|---|
| | 1 | How many digits in a yellow background are there ? | two |
| | 2 | How many digits with a stroke are there among them ? | two |
| | 3 | How many 5's are there among them ? | one |
| | 4 | What is the color of it ? | violet |
| | 5 | What is the style of the digit at the left of it ? | flat |
| | 6 | What is the number of the digit ? | 1 |
| | 7 | What is the background color of the digit ? | white |
| | 8 | What is the number of the digit at the left of it ? | 1 |

Figure 1: Qualitative results of proposed model. Given input image and dialog at top row, retrieved attention (column 1), final attention (column 2) and predicted answers are presented for last three questions. Retrieved attentions focus on reference of ambiguous expressions and final attentions focus on region of target object based on relationship in question. Additionally, we manually modify retrieved attention by assigning high probability to randomly chosen single location (column 3) and show final attention obtained with modified retrieved attention (column 4). With modified retrieved attentions, final attentions also change accordingly. Note that misaligned retrieved attentions are corrected in final attention as depicted in Q8.

⋮

**History:**
How many 6's are there among them ?                     one
What is the number of the digit at the right of it ?    0
What is the style of it ?                               flat
What is the style of the digit at the left of it ?      stroke

**Current QA:** What is the color of the digit ?         violet

| Input image | Retrieved attention from memory | Final attention | Manually modified retrieved attention | Final attention |
|---|---|---|---|---|

Predicted answer: violet                Predicted answer: blue

⋮

**History:**
How many violet digits are there among them ?           two
Are there 6's among them ?                              one
what is the background color of the digit below it ?    salmon
what is the number of the digit below it ?             0

**Current QA:** What is the color of the digit at the right of it ?    blue

| Input image | Retrieved attention from memory | Final attention | Manually modified retrieved attention | Final attention |
|---|---|---|---|---|

Predicted answer: blue                  Predicted answer: red

⋮

**History:**
what is the style of it ?                              flat
what is the color of the 9 ?                           violet
what is the style of the digit below it ?              stroke
what is the background color of the digit ?            silver

**Current QA:** What is the number of the digit ?        5

| Input image | Retrieved attention from memory | Final attention | Manually modified retrieved attention | Final attention |
|---|---|---|---|---|

Predicted answer: 5                     Predicted answer: 8

Figure 2: More Qualitative results of proposed model. Given dialog history, current QA pair (top) and input image (column 1), retrieved attention (column 2), final attention (column 3) and predicted answers are presented in each figure. Manually modified retrieved attention and its corresponding final attention are presented in column 4 and column 5 demonstrating use of retrieved attention.

## B  Qualitative Results on Visual Dialog (VisDial)

Input image

Caption:
*Cat sitting in small bowl on wood flooring indoors*

| QA pair, predicted answer and rank of GT answer | Attended image |
| --- | --- |

**Q1**: *What color is the bowl ?*

GT answer: *White*
Predicted answer: *White*
Rank of GT: *1*

**Q2**: *Do you see any people?*

GT answer: *No*
Predicted answer: *No, just the cat*
Rank of GT: *2*

**Q3**: *What color is the cat ?*

GT answer: *Grey, white, and black*
Predicted answer: *Grey, black and white*
Rank of GT: *6*

**Q4**: *Is the cat wearing any collar?*

GT answer: *No*
Predicted answer: *No*
Rank of GT: *1*

Figure 3: Qualitative results of dialog in VisDial. Given the image and the caption at the top, a sequence of questions are presented in each row with the attention from the proposed model, the GT answer, the predicted answer of the model and the rank of the GT answer. The attention map is concentrated on the reference of question while it is distributed over the entire image when the reference of the question is not present in the image as in Q2 and Q4.

Input image

Caption:
*A hot dog covered in mustard and cheese sits next to French fries*

---

QA pair, predicted answer and rank of GT answer | Attended image

**Q1**: *Are hot dogs in bun ?*

GT answer: *Yes, there's only one though*
Predicted answer: *Yes*
Rank of GT: *9*

**Q2**: *Are they on plate ?*

GT answer: *No, it's in cardboard container*
Predicted answer: *No, it's in cardboard container*
Rank of GT: *1*

**Q3**: *Are they stake fries ?*

GT answer: *No, they are shoestring fries*
Predicted answer: *No, they are shoestring fries*
Rank of GT: *1*

**Q4**: *Are they on table ?*

GT answer: *They seem to be*
Predicted answer: *Yes*
Rank of GT: *17*

Figure 4: Qualitative results of another dialog in VisDial.

| Dialog Information | Input image | Attended image |
|---|---|---|
| Caption: *A large bear standing upright with mountains in the background*<br>Previous QA: *Is this the only bear here ? / yes*<br>Current question: *What color is it's fur ?*<br><br>GT answer: *Brown*<br>Predicted answer: *Brown*<br>Rank of GT: *1* | | |
| Caption: *A train that is on a large rail way*<br>Previous QA: *Is the train moving ? / No it is stopped*<br>Current question: *What color is the train ?*<br><br>GT answer: *It is white and red with some blue on it*<br>Predicted answer: *It is white and red with some blue on it*<br>Rank of GT: *1* | | |
| Caption: *An airplane parked in the middle of a runway*<br>Previous QA: *Can you see the airport ? / No*<br>Current question: *Is it a sunny day ?*<br><br>GT answer: *Yes*<br>Predicted answer: *Yes*<br>Rank of GT: *1* | | |
| Caption: *A zebra standing next to a wire beside a chain link fence*<br>Previous QA: *Is the zebra in a zoo ? / Maybe, can't tell*<br>Current question: *Is the zebra young or old ?*<br><br>GT answer: *Grown I think*<br>Predicted answer: *Young*<br>Rank of GT: *5* | | |

Figure 5: More qualitative results of questions in different dialogs.