[Reviews · NeurIPS 2017]

Reviewer 1



Overall Impressions: I think this is a solid paper. The problem is timely, the paper is well written, the approach is relatively novel and well motivated, and both qualitative and quantitative results are impressive. There are some things I have questions about but these are largely a matter of curiosity and not critique. Strengths: - Writing is relatively clear and the figures do a good job of supporting the text. - The approach is well described and motivated. - The synthetic dataset seems well constructed for this task; the visual perception portion being easy to solve compared to the referential ambiguity. - The various ablations presented in the synthetic experiments were interesting. - Results on the synthetic and Visual Dialog datasets are convincing. Weaknesses: - I would have liked to see some analysis about the distribution of the addressing coefficients (Betas) with and without the bias towards sequential addressing. This difference seems to be very important for the synthetic task (likely because each question is based on the answer set of the previous one). Also I don't think the value of the trade-off parameter (Theta) was ever mentioned. What was it and how was it selected? If instead of a soft attention, the attention from the previous question was simply used, how would that baseline perform? - Towards the same point, to what degree does the sequential bias affect the VisDial results? - Minor complaint. There are a lot of footnotes which can be distracting, although I understand they are good for conserving clarity / space while still providing useful details. - Does the dynamic weight prediction seem to identify a handful of modes depending on the question being asked? Analysis of these weights (perhaps tSNE colored by question type) would be interesting. - It would have been interesting to see not only the retrieved and final attentions but also the tentative attention maps in the qualitative figures.

Reviewer 2



This paper proposed a visual reference resolution model for visual dialog. The authors proposed to attentions, 1: tentative attention that only consider current question and history, and 2: relevant attention that retrieved from an associate attention memory. Two attentions are further combined with a dynamic parameter layer from [9] and predict the final attention on image. The authors create MNIST Dialog synthetic dataset which model the visual reference resolution of ambiguous expressions. The proposed method outperform baseline with large margin. The authors also perform experiments on visual dialog dataset, and show improvements over the previous methods. [Paper Strengths] 1: Visual reference resolution is a nice and intuitive idea on visual dialog dataset. 2: MNIST Dialog synthetic dataset is a plus. 3: Paper is well written and easy to follow. [Paper Weaknesses] My major concerns about this paper is the experiment on visual dialog dataset. The authors only show the proposed model's performance on discriminative setting without any ablation studies. There is not enough experiment result to show how the proposed model works on the real dataset. If possible, please answer my following questions in the rebuttal. 1: The authors claim their model can achieve superior performance having significantly fewer parameters than baseline [1]. This is mainly achieved by using a much smaller word embedding size and LSTM size. To me, it could be authors in [1] just test model with standard parameter setting. To backup this claim, is there any improvements when the proposed model use larger word embedding, and LSTM parameters? 2: There are two test settings in visual dialog, while the Table 1 only shows the result on discriminative setting. It's known that discriminative setting can not apply on real applications, what is the result on generative setting? 3: To further backup the proposed visual reference resolution model works in real dataset, please also conduct ablation study on visDial dataset. One experiment I'm really interested is the performance of ATT(+H) (in figure 4 left). What is the result if the proposed model didn't consider the relevant attention retrieval from the attention memory.